# Hexavalent Chromium Targets Securin to Drive Numerical Chromosome Instability in Human Lung Cells

**DOI:** 10.3390/ijms25010256

**Published:** 2023-12-23

**Authors:** Jennifer H. Toyoda, Julieta Martino, Rachel M. Speer, Idoia Meaza, Haiyan Lu, Aggie R. Williams, Alicia M. Bolt, Joseph Calvin Kouokam, Abou El-Makarim Aboueissa, John Pierce Wise

**Affiliations:** 1Wise Laboratory for Environmental and Genetic Toxicology, University of Louisville, 500 S Preston Street, Building 55A, Room 1422, Louisville, KY 40292, USArspeer@salud.unm.edu (R.M.S.); haiyan.lu@louisville.edu (H.L.); joseph.kouokam@louisville.edu (J.C.K.); 2Department of Pharmaceutical Sciences, The University of New Mexico College of Pharmacy, Albuquerque, NM 87131, USA; ambolt@salud.unm.edu; 3Department of Mathematics and Statistics, University of Southern Maine, Portland, ME 04104, USA

**Keywords:** hexavalent chromium, chromosome instability, centrosome amplification, securin

## Abstract

Hexavalent chromium [Cr(VI)] is a known human lung carcinogen with widespread exposure in environmental and occupational settings. Despite well-known cancer risks, the molecular mechanisms of Cr(VI)-induced carcinogenesis are not well understood, but a major driver of Cr(VI) carcinogenesis is chromosome instability. Previously, we reported Cr(VI) induced numerical chromosome instability, premature centriole disengagement, centrosome amplification, premature centromere division, and spindle assembly checkpoint bypass. A key regulator of these events is securin, which acts by regulating the cleavage ability of separase. Thus, in this study we investigated securin disruption by Cr(VI) exposure. We exposed human lung cells to a particulate Cr(VI) compound, zinc chromate, for acute (24 h) and prolonged (120 h) time points. We found prolonged Cr(VI) exposure caused marked decrease in securin levels and function. After prolonged exposure at the highest concentration, securin protein levels were decreased to 15.3% of control cells, while securin mRNA quantification was 7.9% relative to control cells. Additionally, loss of securin function led to increased separase activity manifested as enhanced cleavage of separase substrates; separase, kendrin, and SCC1. These data show securin is targeted by prolonged Cr(VI) exposure in human lung cells. Thus, a new mechanistic model for Cr(VI)-induced carcinogenesis emerges with centrosome and centromere disruption as key components of numerical chromosome instability, a key driver in Cr(VI) carcinogenesis.

## 1. Introduction

Particulate hexavalent chromium [Cr(VI)] is a potent lung carcinogen with widespread occupational and environmental exposure risks recognized by prominent placement on the Substance Priority List [1,2]. Cr(VI) primarily targets the respiratory system by inhalation and causes cancers of the lungs, nose, and nasal sinuses which have been abundantly observed after occupational exposure [3,4,5,6,7,8,9,10]. Potential nonoccupational Cr(VI) exposure is a concern in industrial areas. In the United States, industrial sources release up to 2900 tons of chromium into the atmosphere and total global emissions are estimated at 58,000 to 112,000 tons annually, though effects of low, chronic levels of airborne Cr(VI) are relatively unknown [11,12]. Studies of Cr(VI) carcinogenicity by ingestion are limited, although cancers of the pharynx, larynx, and stomach have been reported [4,11,13,14,15,16,17,18]. Dermal exposure is known to cause allergic contact dermatitis, but serious injury by this route is rare [19,20]. Cr(VI) can also lead to disorders beyond the port of entry, including the urinary system, hematopoietic system, and reproductive system, and Cr(VI) exposure correlates with Cr measurements in blood, plasma, and urine [11,14,18,21,22,23,24,25,26].

Despite well-known cancer risks, the carcinogenic mechanisms of Cr(VI) are not well understood. A driving mechanism in Cr(VI) carcinogenesis is chromosome instability [27,28,29,30,31]. We previously reported particulate Cr(VI) induces numerical chromosome instability in human lung cells after prolonged (48–120 h) exposures [32,33]. We also found Cr(VI)-induced numerical chromosome instability was heritable at a cellular level and, consistent with our previous reports, requires at least 48 h of exposure to occur [34]. How Cr(VI) causes numerical chromosome instability is currently unknown.

A central regulator of numerical chromosome instability is securin [35,36,37]. Securin maintains numerical chromosome fidelity through multiple mechanisms which include control of centrosome duplication [38,39,40] and centriole disengagement [39,40,41]. Maintaining normal centrosome numbers is key for normal mitosis, as centrosome amplification drives numerical chromosome instability by generating aberrant spindle poles, kinetochore–microtubule disruption, and asymmetrical chromatid segregation [42,43,44,45,46]. In addition, securin is well described as an anaphase inhibitor that regulates separase activity at centromeres via inhibitory binding [36,46,47,48,49]. Dysregulated separase cleavage at centromeres causes premature centromere division and spindle assembly checkpoint (SAC) bypass. Thus, securin protects faithful chromosome segregation through both centromere- and centrosome-localized activities.

We previously demonstrated prolonged Cr(VI) exposure induced premature sister chromatid separation, premature centriole disengagement, and centrosome amplification [29,32,50,51]. These defects can lead to numerical chromosome instability, but their underlying cause following Cr(VI) exposure is unknown. Securin is vulnerable to DNA-damaging agents [35,52] and thus, may be involved in Cr(VI)-induced centrosome amplification and chromosome instability. Securin dysregulation has been observed in several cancers [53]; however, the impacts of Cr(VI) exposure on securin are not understood. This study investigates the hypothesis that securin is targeted by Cr(VI), which leads to numerical chromosome instability.

## 2. Results

### 2.1. Prolonged Cr(VI) Exposure Causes Loss of Securin Protein 

Our earlier studies showed while acute (24 h) Cr(VI) exposure did not induce changes in centrosomes or chromosome instability, prolonged (120 h) exposure induced centrosome amplification, abnormal centrioles, and numerical chromosome instability [29,32,33,50]. Given that securin is a central regulator of centrosome maintenance and prevents abnormal centriole splitting, we investigated whether securin protein levels are affected by acute or prolonged Cr(VI) exposure. We found 24 h exposure caused slightly increased securin levels, and cells exposed for 120 h to concentrations of 0.1, 0.2, and 0.3 μg/cm^2^ zinc chromate showed reductions in securin levels to 50.7%, 32.3%, and 15.3%, respectively, relative to untreated controls (Figure 1). 

### 2.2. Prolonged Cr(VI) Exposure Targets Securin Transcription

In considering how Cr(VI) reduces securin protein levels, one possibility is that Cr(VI) induces protein degradation. In normal cells, securin is degraded via the anaphase-promoting complex/cyclosome (APC/C) during anaphase, allowing separase activation [40,46,47]. To measure securin protein degradation rates, we performed a cycloheximide chase assay after 24, 72, and 120 h zinc chromate exposure. We focused on 0.2 μg/cm^2^ zinc chromate exposure because it shows a strong response in securin reduction after 120 h. We added the intermediate time point of 72 h to detect potential key events in degradation leading up to the observed 120 h reduction in securin protein level. Securin protein half-life did not differ between control and treated cells after 24, 72, or 120 h zinc chromate exposure (Figure 2).

Our data show securin loss is not due to Cr(VI)-induced protein degradation, thus we measured Cr(VI) effects on securin transcription by quantifying securin mRNA using RT-qPCR after 24 and 120 h zinc chromate exposures. Slight decreases in mRNA levels were observed after 24 h. After 120 h, 0.1, 0.2, and 0.3 μg/cm^2^ zinc chromate exposures caused relative securin mRNA expression fold changes of 0.52, 0.143, and 0.079, respectively (Figure 3). Together, these data show Cr(VI) targets securin via pretranslational mechanisms and not by protein degradation.

### 2.3. Prolonged Cr(VI) Exposure Causes Loss of Securin Function

The major function of securin is to regulate the cleavage functions of separase, which it regulates via inhibitory binding, thus preventing untimely activation of separase function during mitosis [54,55,56]. To assess securin function under reduced protein levels following Cr(VI) exposure, we investigated three measures of separase activity: (1) separase autocleavage, (2) kendrin cleavage, and (3) SCC1 cleavage. When securin is degraded and separase becomes free of securin binding, separase becomes its own substrate and cleaves itself [57]. Separase autocleavage is observable on Western blots as a full-length 220 kDa band and lower molecular weight bands corresponding to protein fragments (Figure 4A). After 24 h, separase 220, 170, and 70 kDa bands were similar to control levels at all zinc chromate concentrations we tested (Figure 4B–D). However, 120 h exposure caused the 170 kDa protein fragment to increase to 200%, 310%, and 380% of control levels with 0.1, 0.2, and 0.3 μg/cm^2^ zinc chromate concentrations, respectively (Figure 4C). Separase fragments of 70 kDa increased to 220%, 420%, and 430% of control levels, respectively (Figure 4D). Full-length separase levels increased to 115.4%, 157.7%, and 171.5% above control values, respectively (Figure 4B), indicating cleavage fragments are disproportionately increased relative to full-length separase protein after prolonged exposure. These data indicate separase autocleavage increases with prolonged Cr(VI) exposure, consistent with the loss of securin at the same zinc chromate concentrations.

To rule out that increased cleavage products are a consequence of increased separase expression, we measured separase mRNA levels after 24 and 120 h of zinc chromate exposure. RT-qPCR results show separase mRNA levels were reduced at all time points and concentrations of zinc chromate exposure (Figure 4E). After 24 h exposure to 0.1, 0.2, and 0.3 μg/cm^2^ zinc chromate separase expression was reduced by fold change of 0.761, 0.683, and 0.564, respectively, relative to controls, versus 0.412, 0.145, and 0.085 relative to control levels at 120 h, respectively, indicating downregulation. Thus, the observed increase in separase cleavage products is not likely caused by increased separase protein expression and is most likely due to the loss of its inhibitory partner securin.

Securin functions to control separase-mediated cleavage of multiple mitotic proteins. Among these is kendrin, a large coiled-coil protein that supports centriole engagement at the centrosome [58,59]. Centrioles must remain engaged throughout the cell cycle until anaphase when they split for the purpose of centrosome duplication. Aberrant centriole separation during interphase can cause overduplication of centrosomes and lead to centrosome amplification [38,60,61]. Kendrin cleavage can be observed by Western blot as a full-length band of 360 kDa and a 125 kDa N-terminal fragment (Figure 5A). To assess securin function following Cr(VI) exposure, we measured full-length and cleaved kendrin protein levels. Results showed 24 h exposure induced slight increases in full-length protein, but 120 h exposure to 0.1, 0.2, and 0.3 μg/cm^2^ zinc chromate resulted in full-length protein levels of 75.7%, 56.4%, and 55.8% of control levels, respectively (Figure 5B). Cleaved protein at 120 h increased to 118.3%, 159.9%, and 152.2% of control levels, respectively (Figure 5C). Thus, 120 h zinc chromate exposure resulted in higher cleaved versus full-length kendrin levels, indicating increased separase activity and decreased securin function.

Securin also functions to inhibit separase from cleaving the SCC1 subunit of cohesin. Cohesin rings are present at the centrioles to link them together during interphase and prevent premature disengagement [62]. The full-length SCC1 protein appears as a 130 kDa band, while its cleavage product migrates to 90 kDa (Figure 6A). We measured SCC1 cleavage as a ratio of cleaved to full-length protein. We hypothesize that centriole-associated cohesin is cleaved by prematurely active separase during interphase, and since centrosomes reside in the cytoplasm, we used cytoplasmic protein fractions to analyze SCC1 cleavage, but protein levels were too low to be detected by Western blot. We measured SCC1 in nuclear fragments and found prolonged Cr(VI) exposure increased the ratio of cleaved/full-length protein (Figure 6B). Though the result was not statistically significant, 120 h exposure to 0.1, 0.2, and 0.3 μg/cm^2^ zinc chromate increased the ratio of cleaved/full-length protein to 102.9%, 148.1%, and 152.0% compared to control, respectively. This effect indicates a trend toward increased cleavage of SCC1 by separase and reflects abnormal securin function.

We previously demonstrated zinc chromate induces centriole disruption, including abnormal centriole disengagement in the G2 phase [50]. In the current study, we showed that kendrin and cohesin are cleaved following prolonged Cr(VI) exposure. Both proteins are responsible for maintaining centriole engagement throughout the cell cycle, until timely release in late mitosis or the early G1 phase [63]. Our current results showing increased separase autocleavage (i.e., activity) and kendrin and cohesin cleavage are consistent with our previously published data and strongly support a conclusion that Cr(VI) targets centriole linker maintenance.

### 2.4. Prolonged Cr(VI) Exposure Compromises Compensatory Separase Regulators

While securin is the main regulator of separase, its function is protected by compensatory mechanisms in the cell. Studies show separase control can be retained even after securin knockout [64,65], leading to questions of how Cr(VI) might circumvent compensation for securin loss. Cyclin B1 acts as a secondary separase inhibitor [54,66] by activating cyclin-dependent kinase 1 (CDK1)-dependent phosphorylation of separase, which inactivates it by promoting its aggregation and precipitation [54]. Additionally, the CDK1-cyclin B1 complex binds to separase, which also renders it inactive but primed for activation [54]. Thus, to further evaluate separase regulation, we measured cyclin B1 protein levels after 24 and 120 h of 0.1, 0.2, and 0.3 μg/cm^2^ zinc chromate exposure. No significant change was observed after 24 h exposure, but after prolonged 120 h exposure, cyclin B1 levels were reduced to 43.7%, 24.5%, and 12.7% of control values (Figure 7). These data indicate that Cr(VI) not only reduces securin but also inhibits secondary compensation by cyclin B1. 

Cyclin B1 levels fluctuate with the phases of the cell cycle such that levels are low in G1, begin to rise in S, peak in early mitosis, and drop with its degradation at the metaphase-to-anaphase transition [54,67]. Securin levels rise during S phase and are also depleted via degradation during the initiation of anaphase. One interpretation of the observed reduction in securin and cyclin B1 following Cr(VI) exposure is that Cr(VI) may affect the cell cycle in a manner that leads to accumulation of cells in G1 phase. To explore this hypothesis, we performed cell cycle analysis by flow cytometry and measured cell cycle phases using DNA content. Cr(VI) exposure for 24 and 120 h did not enrich the population of G1 cells. In fact, consistent with a DNA damage exposure, Cr(VI) caused an increase in the G2/M population (Figure 8). G2 and M are the phases at which securin and cyclin B1 levels should be the highest; hence, their loss following Cr(VI) exposure is not due to accumulation of cells in G1 phase.

## 3. Discussion

Particulate Cr(VI) is a known lung carcinogen as demonstrated through studies in chromate workers, experimental animals, and cell culture [9,68,69,70,71,72], yet its molecular mechanism of carcinogenesis is not fully understood. Chromium is proposed to disrupt cellular processes by two approaches. First, inside the cell Cr(VI) is rapidly reduced to Cr(III) by agents such as ascorbate and glutathione, producing chromium intermediate species and reactive oxygen species that can damage intracellular molecules [73,74,75,76]. Secondly, the ability of Cr(III) to bind to proteins and guanine bases raises the possibility for direct interactions to damage molecular targets [77,78,79]. 

Chromosome instability is a key driving mechanism in Cr(VI) carcinogenesis. It is a hallmark of cancer and an early effect of Cr(VI) exposure [28,30,31,34,80]. Chromosome instability has two main categories: (1) structural chromosome instability, which features chromosome breaks and translocations, and (2) numerical instability characterized by the loss or gain of entire chromosomes [81,82,83]. Cr(VI)-induced structural chromosome damage has been linked to formation of DNA lesions, abasic sites, potential replication fork stalling, and loss of DNA repair mechanisms [84,85,86,87]. How Cr(VI) induces numerical chromosome instability is not clear, but it is known that mitotic disruption can cause asymmetrical segregation of chromosomes resulting in aneuploid daughter cells [88,89,90,91]. This study shows Cr(VI) targets securin, a protein that is central to numerical chromosome instability. 

Our data show that prolonged particulate Cr(VI) exposure targets securin and severely decreases its protein levels. Loss of securin protein would be expected to result in increased separase activity, which we observed as evidenced by increased separase autocleavage. A major role for securin is to keep separase inactive until anaphase where it targets multiple proteins for cleavage [36,49,92,93,94]. For example, securin regulates separase in mitosis as part of the spindle assembly checkpoint [46,95]. It regulates separase cleavage at centromeres to restrict chromatid division until proper spindle assembly and chromosome alignment [46,47,96]. It only releases separase at the precise time needed to cleave the SCC1 subunit of cohesin to allow for centromere separation [46,47,56,63]. We found particulate Cr(VI) increased SCC1 cleavage consistent with the loss of securin and activation of separase. This outcome is also consistent with our previous reports that Cr(VI) induces premature anaphase, premature centrosome separation, and centromere spreading [33,51]. 

Securin also regulates separase to control the timing of centriole disengagement and duplication [59]. It regulates separase activity at centrosomes to restrict centrosome duplication to once per cell cycle [39,40,59,61,62,63]. Disrupting this timing could lead to centrosome amplification resulting in multipolar mitotic division and numerical chromosome instability. We found particulate Cr(VI) increased kendrin cleavage consistent with the loss of securin and activation of separase. This outcome is also consistent with our previous reports that Cr(VI) induces centriole splitting and centrosome amplification [32,33,50]. 

Cells can compensate for the absence of securin with functional redundancy of cyclin B1, which can also restrain separase activity [54,67,97]. It is unclear how much securin loss is necessary to trigger this function of cyclin B1. However, we found prolonged particulate Cr(VI) also targeted this compensatory mechanism and reduced cyclin B1 levels. Thus, cyclin B1 is not available to compensate for the loss of securin, supporting our hypothesis that the targeting of securin is fundamental to Cr(VI)-induced chromosome instability. 

After observing Cr(VI)-induced loss of securin levels and securin function, we sought to determine the mechanism of securin disruption. Particulate Cr(VI) could cause loss of securin by degrading the protein, by arresting cells in G1 phase of the cell cycle, or by preventing its transcription. Protein half-life measurements show particulate Cr(VI)-induced securin loss was not caused by enhanced protein degradation. In addition, we found prolonged particulate Cr(VI) exposure did not induce a G1 arrest, rather, it induced a G2 arrest consistent with published reports showing G2 arrest after acute Cr(VI) exposures [98,99,100,101]. In contrast, RT-qPCR analysis of securin mRNA showed particulate Cr(VI) targets securin by depressing its transcription. 

How Cr(VI) targets transcription is poorly understood and potential explanations are complex. It may be that transcription factors are prevented from binding to the promoter. Cr(VI) and Cr(III) complexes have been reported to both activate and repress nuclear binding of transcription factors, including Sp1 [102,103], an enhancer of securin transcription. Securin has both enhancer and repressive transcription factors. For example, considering a subset of possible transcription factors, securin promoter activity depends on binding of both the Sp1 transcription factor to the nucleotide region −541 to −500 and NF-Y transcription factor binding to the region −519 to −497 [53,104], while direct binding of the repressive factor, KLF6, to the securin promoter was observed between nucleotides −401 to −246 [105]. Thus, it could be specific binding patterns allow the repressive effect to dominate and prevent gene transcription. Changes in DNA methylation could also prevent access to promoter regions. Chromate-induced lung tumors have shown increased methylation at some tumor suppressor genes related to reduced gene products [106,107] and in vitro Cr(VI) exposure increased methylation at p16, which negatively correlated to its expression level in human bronchial cells [108]. Cr(VI) is known to alter chromatin organization at large by multiple mechanisms including rearrangement of nucleosomes which alters transcription [109,110]. Another possible cause of Cr(VI)-induced protein and mRNA loss is microRNA disruption. MicroRNA sequencing showed Cr(VI) caused global miRNA dysregulation, including up- and downregulation of multiple miRNAs involved in several cancer pathways [111]; however, securin was not specifically targeted by any of the altered miRNAs (unpublished data). 

## 4. Materials and Methods

### 4.1. Chemicals and Reagents

DMEM and Ham’s F-12 (DMEM/F-12) 50:50 media, glutagro 200 mM L-alanyl-L-glutamine supplement, sodium pyruvate, and Dulbecco’s phosphate-buffered saline (DPBS), tissue culture flasks, dishes, and plasticware were purchased from Corning, Inc. (Manassas, VA, USA). Cosmic calf serum and penicillin/streptomycin were purchased from HyClone (Logan, UT, USA). Sodium pyruvate (100 mM) and MycoAlert kit were purchased from Lonza (Allendale, NJ, USA). Trypsin-EDTA (0.25%) and KaryoMAX^®^ Colcemid Solution (10 ug/mL) were purchased from Gibco. Zinc chromate (CAS# 13530-65-9, 99.7% purity) was purchased from Pfaltz and Bauer (lot Z00277, Waterbury, CT, USA). HALT protease and phosphatase inhibitor cocktail, RIPA buffer, NE-PER nuclear and cytoplasmic extraction reagents, mirVana miRNA isolation kit, High-Capacity cDNA Reverse Transcription, TaqMan Assays, glass chamber slides, and Super Up Rite slides were purchased from Thermo Fisher Scientific Inc. (Waltham, MA, USA). Mini-Protean TGX gels, 4X protein sample loading buffer, Odyssey blocking buffer, IRDye^®^ 800CW, and IRDye^®^ 680RD near-infrared fluorescent secondary antibodies were purchased from Li-Cor (Lincoln, NE, USA). Tween-20, cycloheximide, and methanol were purchased from VWR International (Radnor, PA, USA). Anti-separase (ab16170) and anti-gamma-tubulin (ab11316) monoclonal mouse antibodies were purchased from Abcam (Eugene, OR, USA). Anti-securin (13445S) rabbit monoclonal antibody was purchased from Cell Signaling Technology (Danvers, MA, USA). Anti-alpha-tubulin (GTX628802) and anti-GAPDH (GTX627408) mouse monoclonal antibodies were purchased from GeneTex (Irvine, CA, USA). Kendrin/pericentrin (A301-348A-M) rabbit antibody was purchased from Bethyl Laboratories (Montgomery, TX, USA). Anti-lamin B1 (33–2000) mouse monoclonal antibody was purchased from Invitrogen/Thermo Fisher (Rockford, IL, USA). Anti-SCC1 monoclonal guinea pig antibody was a gift from Dr. Olaf Stemmann (University of Bayreuth, Bayreuth, Germany).

### 4.2. Cell Culture

The human lung cell line used was WTHBF-6 cells, a bronchial fibroblast cell line developed from normal primary human bronchial fibroblasts purchased from Clonetics (San Diego, CA, USA) and isolated from a 67-year old Caucasian male, as previously described [54]. This clonal cell line has an hTERT-extended lifespan with a normal, stable karyotype and displays the same growth rate and cytotoxic and clastogenic response to metals as the primary parent cells [112]. Using this cell line enables consistent cell passaging and prolonged exposure periods used in our toxicological assays. Fibroblast cell lines are relevant cell models for Cr(VI)-induced lung cancer due to observations that chromium deposits in the bronchial stroma of chromate workers, but not in the epithelium [113]. Human fibroblast cells are typically employed in toxicological assays concerning aneuploidy, given that currently available epithelial cell lines demonstrate high background aneuploidy. Our study focuses on chromosomal instability and thus a stable control karyotype is an important prerequisite.

WTHBF-6 cells were maintained according to our published methods [112] as an adherent, subconfluent layer in DMEM/F-12 media, supplemented with 15% cosmic calf serum, 0.2 mM L-alanyl-L-glutamine, 100 IU/mL penicillin, 100 mg/mL streptomycin, and 0.1 mM sodium pyruvate. Cells were maintained in a humidified incubator at 37 °C and 5% CO_2_. 

### 4.3. Preparation of Zinc Chromate and Cell Treatments

All experiments were performed on logarithmically growing cells. WTHBF-6 cells have a doubling time of approximately 24 h [112] and were allowed 48 h after seeding to enter logarithmic growth phase before beginning treatments. Experiments were performed in 100 mm tissue culture dishes and seeded at cell densities appropriate for durations and concentrations of exposure to result in 70–80% confluency at time of harvest. For protein analysis, protein half-life analysis, and RNA analysis, 500,000 cells were seeded for all 24 h treatments. For 120 h treatments, because of the relative cytotoxicity that occurs and to avoid growth to overconfluence of control cells, 60,000, 130,000, 230,000, and 350,000 cells were seeded for zinc chromate treatments of 0, 0.1, 0.2, and 0.3 μg/cm^2^, respectively, such that at harvest the cells had similar numbers of cells available for analysis. Similarly, for cell cycle analysis, 400,000 cells were seeded for all 24 h exposures, and for 120 h exposures 56,000, 120,000, 200,000, and 280,000 cells were seeded for zinc chromate treatments of 0, 0.1, 0.2, and 0.3 μg/cm^2^, respectively, to ensure no change in cell cycle profiles due to contact inhibition and so that similar numbers of cells were available across concentrations at harvest [33]. According to published methods [114], zinc chromate was prepared by washing twice with deionized H_2_O to remove water-soluble contaminants, rinsed twice with acetone to remove organic contaminants, and thoroughly dried. Washed zinc chromate was suspended in sterile water and stirred overnight at 4 °C. Before treatment, fresh medium was added to cell dishes and zinc chromate suspension was applied at a concentration of 0, 0.1, 0.2, or 0.3 μg/cm^2^, as specified for experiment and cell type. These concentrations treat the cells with 1.6–4.7 ug Cr(VI). Treatment durations were 24 and 120 h. Allowable occupational exposure levels to Cr(VI) have a maximum of 5 ug/m^3^ as a time-weighted average [1]. Levels of Cr(VI) in the general environment have been reported to range from 0.001 to 0.1 ug/m^3^ [1]. Given an average respiration of 23 m^3^ of air per 24 h under normal activity, then environmental exposures would be 0.023 to 2.3 ug Cr(VI) for 24 h and 0.115 to 11.5 ug Cr(VI) for a 120 h. Occupational exposures would be 38.4 ug for one 8 h day and 192 ug for five 8 h days. Thus, our treatments model both environmental and workplace exposures. Cells were maintained during the treatment period at 5% CO_2_ in a humidified incubator at 37 °C with no further passaging or media changes until cell harvest at the end of zinc chromate exposure.

### 4.4. Protein Analysis

Cells were seeded in 100 mm tissue culture dishes and treated with zinc chromate as described above. Cell-equivalent whole cell protein analysis was performed by Western blot as previously published [115]. At the end of the Cr(VI) exposure period, cells were released from dishes by incubation with 0.25% trypsin-EDTA. Cells were collected, washed in PBS, and counted with Beckman Coulter Multisizer 4e. Cells were lysed in RIPA buffer supplemented with 100X Halt protease and phosphatase inhibitor cocktail. Cell lysis was centrifuged 10 min at 14,000 rpm and supernatant was collected and prepared with 4X protein sample loading buffer. Protein samples in loading buffer were heated for 10 min at 70 °C and stored at −20 °C. Prepared sample volumes equivalent to 75,000–81,000 lysed cells (antibody-dependent) were loaded into Mini-Protean TGX gels. After electrophoresis, protein lysates were transferred to 0.45 um nitrocellulose or PVDF membranes in ice-cold transfer buffer with 10–20% methanol (antibody-dependent). Membranes were probed with primary antibodies and secondary near-infrared antibodies. Membranes were scanned using an Odyssey CLx scanner and analyzed with Li-Cor Image Studio software version 2.1.10. 

Cytoplasmic and nuclear protein extractions were performed with NE-PER nuclear and cytoplasmic extraction reagents (Thermo Scientific, Waltham, MA, USA) according to the manufacturer’s instructions. Cells were harvested and counted as described above for whole cell extraction. Cell pellets were resuspended in ice-cold CERI (cytoplasmic extraction reagent I) supplemented with 100X Halt protease and phosphatase inhibitor, vortexed vigorously for 15 s and incubated on ice for 15 min. Ice-cold CERII (cytoplasmic extraction reagent II) was added and the tube was vortexed for 5 s, incubated on ice for 1 min, vortexed for 5 s, and centrifuged at 14,000 RPM for 5 min. Cytoplasmic supernatant was isolated and stored on ice until preparation with 4X protein sample loading buffer as for whole cell protein samples. The nuclear fraction was resuspended in NER (nuclear extraction reagent) supplemented with 100X Halt protease and phosphatase inhibitor. Tubes were vortexed on the highest setting for 15 s every 10 min for a total incubation time of 40 min. Samples were centrifuged at 14,000 RPM and the nuclear fraction was prepared with 4X protein sample loading buffer as for whole cell protein samples and stored at −20 °C.

### 4.5. Protein Half-Life Analysis

Protein half-life analysis was performed as previously published [115] with some modifications. Cells were seeded in 100 mm tissue culture dishes and treated with zinc chromate as described above at the final concentrations of 0 or 0.2 μg/cm^2^. Treatment durations were 24, 72, and 120 h. At the end of the exposure time, cells were treated with cycloheximide at a final concentration of 10 uM. Cells were harvested at 0, 2, 4, 6, and 8 h after cycloheximide addition and whole cell protein was analyzed according to the protein analysis method described above. Protein half-life was calculated from the exponential best fit curve of protein levels after 0, 2, 4, 6, and 8 h cycloheximide using the equation Rate = (LN(0.5))/b, where the best fit line equation is y = ae^bx^.

### 4.6. RNA Extraction, cDNA Synthesis, and RT-qPCR

RNA analysis was performed as previously published [115]. After cell seeding and treatment as described for protein analysis, cells were released from dishes by incubation with 0.25% trypsin-EDTA. Cells were collected and washed with PBS. Total RNA was extracted using the mirVana miRNA isolation kit according to the manufacturer’s instructions. Cells were lysed with lysis binding buffer included in the kit and the lysate was kept on ice and homogenized. RNA was extracted with acid-phenol:chloroform, isolated by glass-fiber filter, and washed with mirVana kit reagents. RNA was eluted from the filter. Total RNA quality and quantity were measured by spectrophotometry using a NanoDrop instrument. Samples were stored at −80 °C. 

Total RNA was reverse transcribed to single-stranded cDNA using a High-Capacity cDNA Reverse Transcription Kit according to the manufacturer’s instructions. The 2X master mix was prepared containing random primers and reverse transcriptase. An equivalent amount of RNA across all experimental conditions, up to the maximum of 2 ug per reaction, was added to the master mix for each sample. Control samples with no reverse transcriptase and no input RNA were included with each experiment. Reverse transcription was performed on a Biometra thermocycler. cDNA was stored at −20 °C for no longer than one week before RT-qPCR analysis.

RT-qPCR was performed using TaqMan™ 20X RNA Assays and TaqMan™ Universal Master Mix II, with UNG, or TaqMan Fast Advanced Master Mix, with UNG. Endogenous mRNA (GAPD) and target mRNA were analyzed in duplex. RT-qPCR reactions were performed in triplicate and controls for no reverse transcriptase, no RNA input, and no cDNA template were included for each of the three independent experiments. Quantitation was performed using a StepOne Plus Real-Time PCR System. Results were normalized by the ∆∆Ct method and expressed as relative quantification compared to untreated control (∆Ct = Ct gene target—Ct endogenous control; ∆∆Ct = ∆Ct sample 1—∆Ct calibrator (untreated control); fold change = 2^−∆∆Ct^). The calibrator has a relative quantification of 1. Relative quantification value of 10 means the gene is 10 times more expressed, while a value of 0.1 means 10 times less expressed.

### 4.7. Cell Cycle Analysis 

Cell cycle analysis was performed by flow cytometry using published methods [56] with modifications. Cells were seeded in 100 mm tissue culture dishes and treated with zinc chromate as described above for 24 h or 120 h. One hour before the end of treatment, neocarzinostatin (400 ng/mL final concentration) was applied to the positive control dish and incubated for 30 min in the dark, followed by 30 min recovery time in fresh media. At the end of the treatment time, the media were collected in 50 mL conical tubes to ensure analysis of all cells. Adherent cells were released from dishes with 0.25% trypsin-EDTA and collected and combined with harvested media. Cells were centrifuged to a pellet at 1000 RPM for 5 min, then resuspended in 5 mL PBS and counted. One million cells for each treatment condition were transferred to a new tube and resuspended in 0.5 mL of PBS. Cells were fixed in 4% PFA (paraformaldehyde) on ice for 15 min. Cells were pelleted by centrifugation, washed with PBS, and centrifuged again. Cells were resuspended in 0.5 mL PBS and combined with 70% ethanol, added dropwise during gentle vortexing. Fixed cells were centrifuged, resuspended in 1 mL 70% ethanol, and stored at −20 °C until analysis. Cell cycle analysis was performed using propidium iodide to quantify DNA content. For each sample, 200,000 cells were analyzed. Flow cytometry data were analyzed using the FlowJo software version 7 to determine the percentage of cells in the G1, S, or G2/M phase.

### 4.8. Cell Authentication 

Karyotype analysis using Giemsa banding (g-banding) was used to confirm cell line identity. G-banding creates signature staining patterns on chromosomes to enable identification of each chromosome and to characterize a cell line. Metaphases were prepared and g-banded on glass microscope slides and imaged using an Applied Spectral Imaging microscope and software version 1.2. Ten metaphases were assessed per analysis. In addition to karyotype analysis, short tandem repeat (STR) analysis was performed annually by American Type Culture Collection (ATCC) to confirm cell line identity. Cells in continuous culture were tested monthly to confirm cultures were mycoplasma negative. 

### 4.9. Statistical Analysis

All values were expressed as the mean ± standard error of the mean. For cell cycle analysis, Dunnett’s multiple comparisons test (α = 0.05) was used to assess differences between treatments and controls in each cell phase. Two-tailed Student’s *t*-test was performed for all other assays to determine differences between each chromate concentration and the untreated control at each time point.

## 5. Conclusions

This study shows Cr(VI) targets securin and compromises securin’s inhibitory function on separase. Considered together with published data, a new mechanistic model emerges for particulate Cr(VI)-induced numerical chromosome instability (Figure 9). Particulate Cr(VI) targets securin transcription resulting in loss of securin protein leading to loss of separase inhibition. The inappropriate separase activity results in elevated cleavage of its substrates, including centriole linkers and centromeres. The inappropriate centriole cleavage leads to centriole disengagement and promotes centrosome overduplication resulting in centrosome amplification, while the inappropriate cleavage at centromeres results in spindle assembly checkpoint bypass leading to premature anaphase, premature centrosome separation, and centromere spreading. The centrosome amplification and the spindle assembly checkpoint bypass then drive numerical chromosome instability, which confers genotypic plasticity advantageous to Cr(VI)-induced transformation and carcinogenesis. Our future course will be to consider how Cr(VI) targets securin transcription considering Cr binding to promoter regions, impacting the relative amounts and binding of transcription factors and possible epigenetic changes.

## Figures and Tables

**Figure 1 ijms-25-00256-f001:**
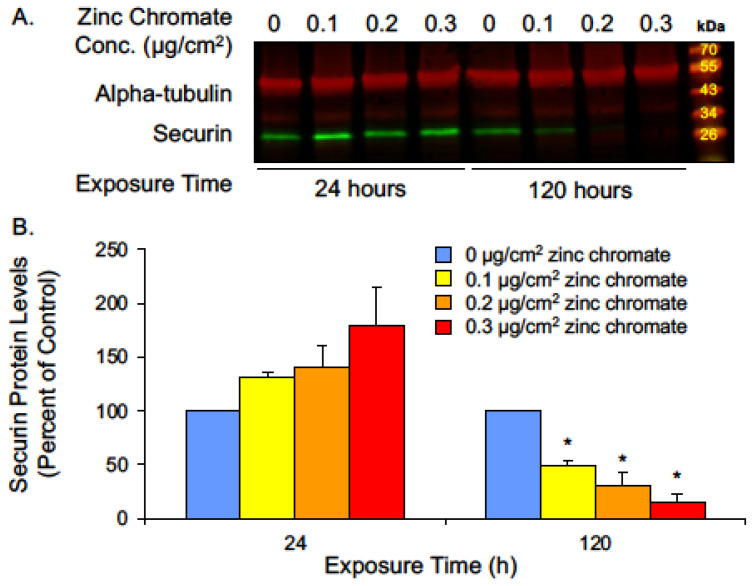
Prolonged Cr(VI) exposure decreases securin protein levels. (**A**) Representative Western blot for securin. Alpha-tubulin was used as a loading control. (**B**) Securin whole cell protein levels decreased after 120 h Cr(VI) exposure. Data are expressed as percent of untreated control cells and reflect the mean of three independent experiments. Error bar = standard error of the mean. * Significantly different from the control group (*p* < 0.05).

**Figure 2 ijms-25-00256-f002:**
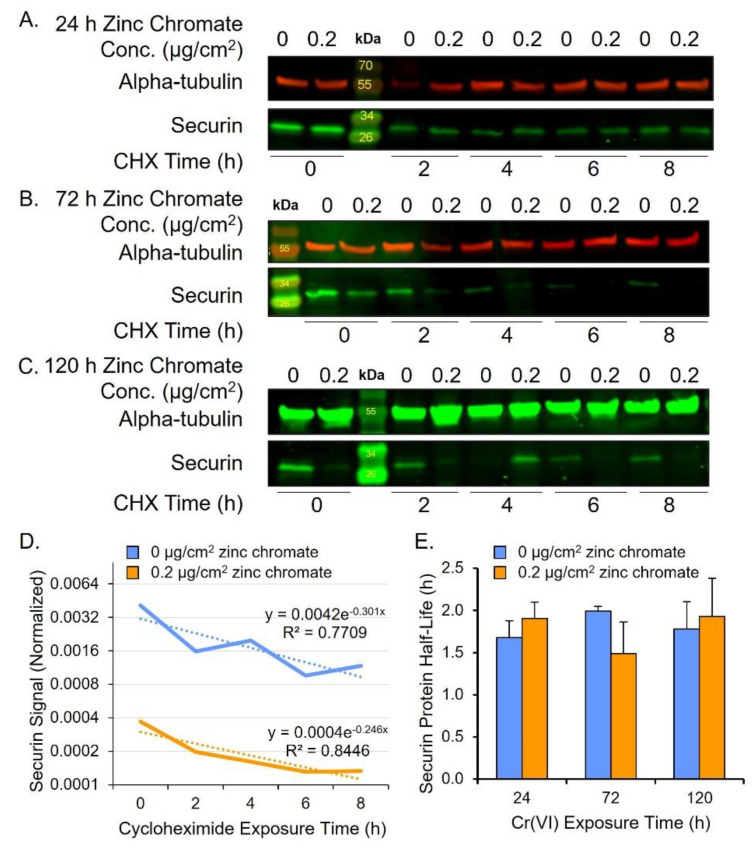
Cr(VI) does not change rates of securin degradation. (**A**–**C**) Representative Western blots for securin after cycloheximide (CHX) treatment. Alpha-tubulin was used as a loading control. (**A**) Securin protein after 24 h Cr(VI). (**B**) Securin protein after 72 h Cr(VI). (**C**) Securin protein after 120 h Cr(VI). (**D**) Representative plot of securin protein degradation over time on log base 2 scale (120 h zinc chromate exposure example). Solid lines represent the securin signal and dotted lines represent the best fit line used for calculating the protein half-life. (**E**) Securin half-life did not significantly change after zinc chromate exposure. Data reflect the mean of three independent experiments. Error bar = standard error of the mean. No condition was significantly different from the untreated control group.

**Figure 3 ijms-25-00256-f003:**
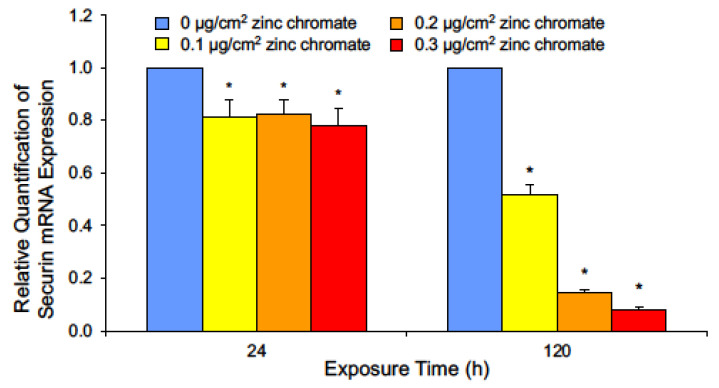
Cr(VI) exposure decreases securin mRNA after 24 and 120 h. Data are expressed as relative expression compared to untreated control cells and reflect the mean of three independent experiments with three technical replicates each. Error bar = standard error of the mean. * Significantly different from the control group (*p* < 0.05).

**Figure 4 ijms-25-00256-f004:**
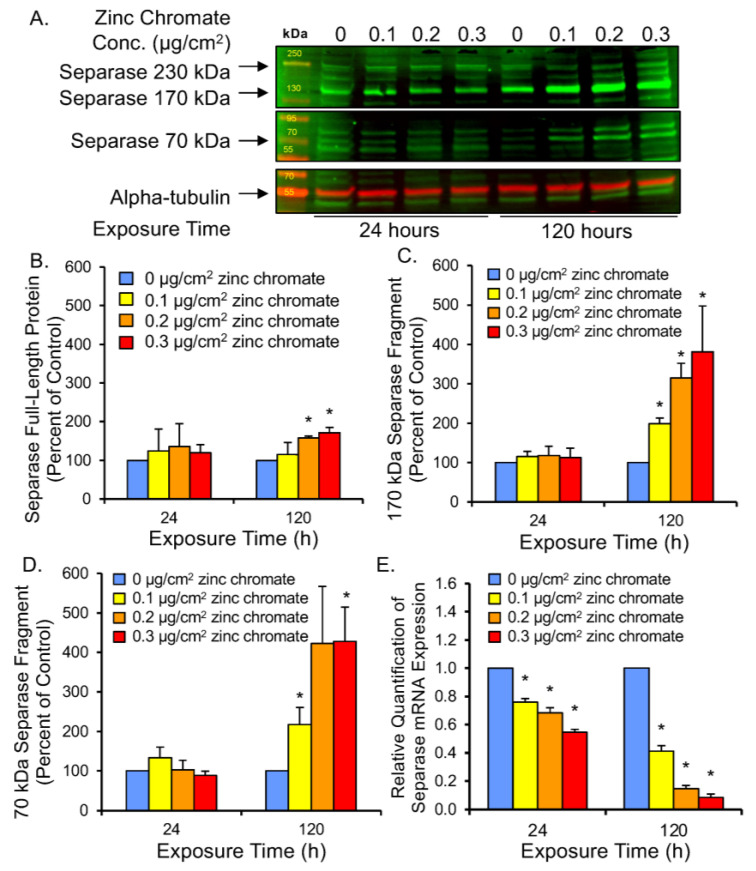
Prolonged Cr(VI) exposure induced separase cleavage and decreased separase mRNA expression. (**A**) Representative Western blot for separase. Alpha-tubulin was used as a loading control. (**B**) Separase full-length protein levels increased slightly after 120 h Cr(VI). (**C**,**D**) Cleaved separase protein levels greatly increased after 120 h Cr(VI). (**E**) Separase mRNA levels decreased after 24 and 120 h zinc chromate exposure. Data are expressed as percent of untreated control cells and reflect the mean of three independent experiments. Error bar = standard error of the mean. * Significantly different from the control group (*p* < 0.05).

**Figure 5 ijms-25-00256-f005:**
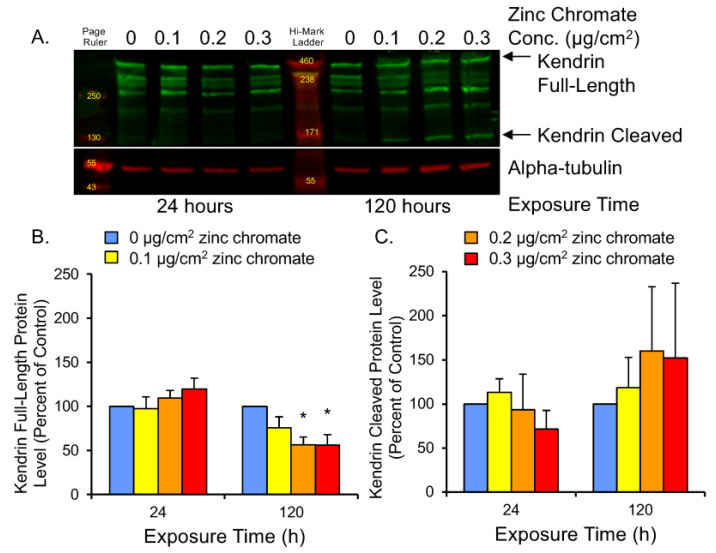
Prolonged Cr(VI) exposure induced kendrin cleavage. (**A**) Representative Western blot showing full-length and cleaved kendrin bands. Alpha-tubulin was used as a loading control. (**B**) Full-length kendrin levels decreased after 120 h Cr(VI). (**C**) Cleaved kendrin protein levels increased after 120 h. Data are expressed as percent of untreated control cells and reflect the mean of three independent experiments. Error bar = standard error of the mean. * Significantly different from the control group (*p* < 0.05).

**Figure 6 ijms-25-00256-f006:**
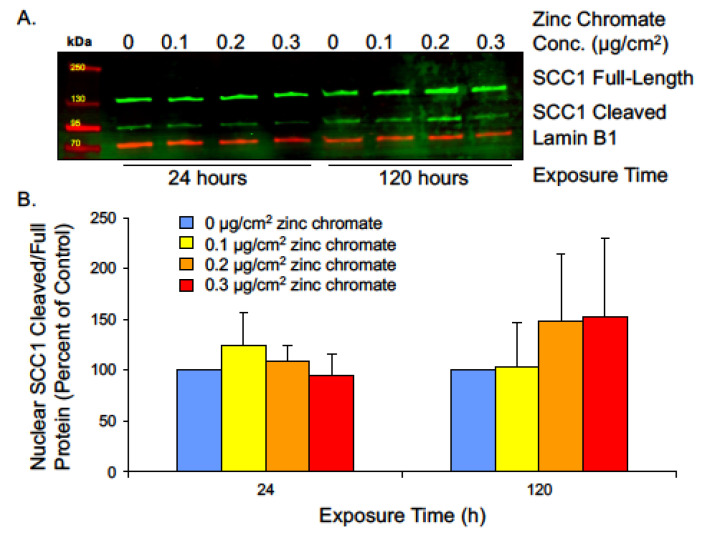
Prolonged Cr(VI) exposure increases nuclear SCC1 cleavage. (**A**) Representative Western blot showing full-length and cleaved SCC1 bands from nuclear extract. Lamin B1 was used as a loading control. (**B**) The ratio of cleaved/full-length nuclear SCC1 was not significantly altered but showed an increasing trend, indicating increased cohesin cleavage. Data are expressed as percentage of untreated control cells and reflect the mean of two independent experiments. Error bars = standard error of the mean. No condition was significantly different from the control group.

**Figure 7 ijms-25-00256-f007:**
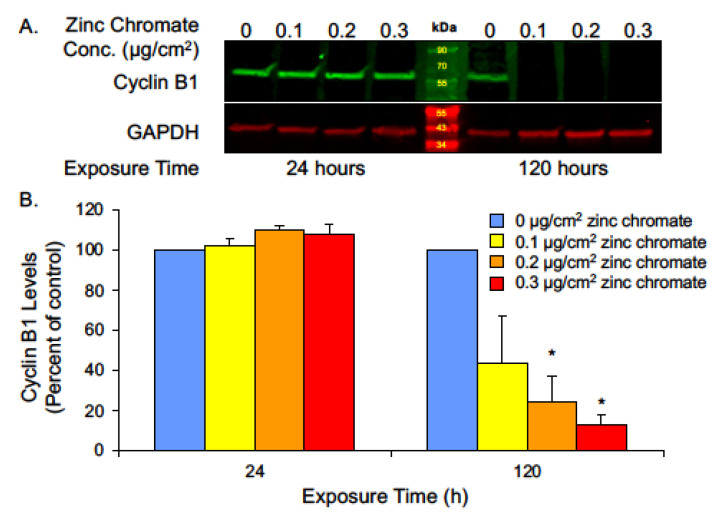
Prolonged Cr(VI) exposure decreased cyclin B1 levels. (**A**) Representative Western blot for cyclin B1. GAPDH was used as a loading control. (**B**) Cyclin B1 whole cell protein levels decreased after 120 h Cr(VI) exposure. Data are expressed as percent of untreated control cells and reflect the mean of three independent experiments. Error bar = standard error of the mean. * Significantly different from the control group (*p* < 0.05).

**Figure 8 ijms-25-00256-f008:**
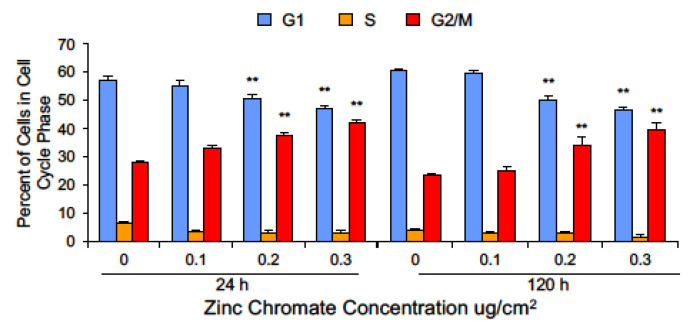
Cr(VI) decreased the percentage of cells in G1 and increased the percentage of cells in G2/M. Data are expressed as percent of untreated control cells and reflect the mean of three or four independent experiments. Error bar = standard error of the mean. ** Significantly different from the control group (*p* < 0.001).

**Figure 9 ijms-25-00256-f009:**
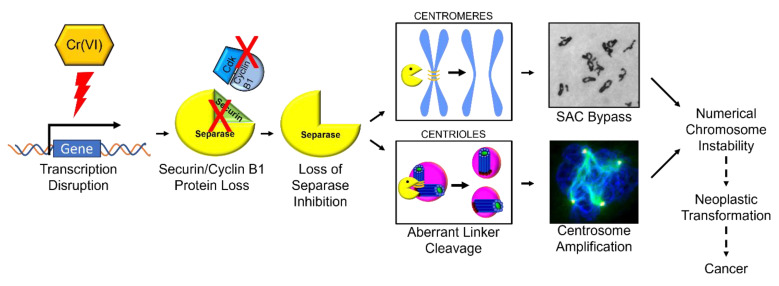
Mechanistic model of Cr(VI)-induced numerical chromosome instability.

## Data Availability

The data presented in this study are available on request from the corresponding author. The data are not publicly available due to privacy limitations.

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
