# Peer review of "Hexavalent Chromium Targets Securin to Drive Numerical Chromosome Instability in Human Lung Cells"

_ijms, 2023, doi:10.3390/ijms25010256_

Round 1

Reviewer 1 Report

Comments and Suggestions for Authors

Please find the following comments.

1) What is the cell seeding density and how about the cell confluency after 120h exposure? 

Was the cells passaged or was medium changed during 120h exposure? These information should be provided.

2) Effects of exposure for 24h and 120h on cell viablity change or cell proliferation should be examined. Cell viability/proliferation assay such as MTS assay can be carried out.

3) The data of 24h and 120h were normalized to 24h or 120h exposed 0 ug group. This will induce loss of the information of comparasion between 24h and 120h. Therefore, this nomalization seems to be not necessary.

Reviewer 2 Report

Comments and Suggestions for Authors

Dear Authors,

Thank you very much for submitting such an interesting paper to the International Journal of Molecular Sciences

Below I present several minor comments and suggestions:

- please correct the references section according to the guidelines for the IJMS journal

- Line 39, it would be good to provide examples of other cancers that might occur due to hexavalent chromium exposure, just as an example for the readers

- In the introduction, could you provide examples in which hexavalent chromium might be absorbed by the body? is it only through the inhalation or are there any other routes?

- the titles of the separate paragraphs should be numbered. There is no need to use capital letters. instead of INTRODUCTION, use '1. Introduction'. Correct other titles in the manuscript accordingly.

- regarding figure 3 - I am aware that using bright colors is very beneficial for this kind of schema however these colors are a little bit too bright. What about choosing more toned colors? Just a minor suggestion

- Line 446, it would be good to add information on what is known about the molecular mechanisms of hexavalent chromium-induced carcinogenesis. Just as examples of what is currently known before going with further discussion

- Line 519 - it can be a separate section entitled 'Conclusions'

Kind regards

Round 2

Reviewer 1 Report

Comments and Suggestions for Authors

The authors have responded properly to most of the reviewer's comments. Please pay attention to the following additional comments:

1) Line 141-144. “Optimal cell seeding densities are determined by previous cytotoxicity analyses which show exposure to 0.1, 0.15, and 0.2 uc/cm2 zinc chromate for 24 h lead to 76%, 64%, and 53% relative survival while 120 h exposure resulted in 36%, 24%, and 16% relative survival, respectively [33].”

Comments:

The doses of zinc chromate used in this study seems to be extremely high that resulted very low survival rate, especially in 120h exposure. Is the low survival rate due to cell death such as apoptosis/necrosis, or due to inhibition on cell proliferation? Because we understand that under cell death, a lot of signaling pathway will change. It is suggested to add some discussion on the doses used in this study. Are these doses responsible for carcinogenesis? Because if these doses are too high and lethal, the underlying mechanism may be not responsible for carcinogenesis, but cell death.

And if it is possible, it is recommended to include lower doses into the experiment design and compare the changes with doses used in the study.

2) for the response to comments 4) The data of 24h and 120h were normalized to 24h or 120h exposed 0 ug group. This will induce loss of the information of comparison between 24h and 120h. Therefore, this normalization seems to be not necessary.

Authors' response: The authors have decided to represent the data as relative to the exposure controls for the following reasons.

Comments: especially considering that 24h and 120h exposure induced different survival rate, keep the information by undo the unnecessary normalization will help interpret the results.

3) The authors revised the manuscript by deleting a whole paragraph. However, this makes it difficult to recognize what are the modified parts. The unchanged sentences should not be deleted. Changes should be made only to the necessary places.
